# Learning a Latent Search Space for Routing Problems using Variational Autoencoders

**André Hottung**
Bielefeld University
Bielefeld, Germany
`andre.hottung@uni-bielefeld.de`

**Bhanu Bhandari**
University of Massachusetts Amherst
Amherst, MA, USA
`bhanubhandar@cs.umass.edu`

**Kevin Tierney**
Bielefeld University
Bielefeld, Germany
`kevin.tierney@uni-bielefeld.de`

## Abstract

Methods for automatically learning to solve routing problems are rapidly improving in performance. While most of these methods excel at generating solutions quickly, they are unable to effectively utilize longer run times because they lack a sophisticated search component. We present a learning-based optimization approach that allows a guided search in the distribution of high-quality solutions for a problem instance. More precisely, our method uses a conditional variational autoencoder that learns to map points in a continuous (latent) search space to high-quality, instance-specific routing problem solutions. The learned space can then be searched by any unconstrained continuous optimization method. We show that even using a standard differential evolution search strategy our approach is able to outperform existing purely machine learning based approaches.

## 1 Introduction

Significant progress has been made in learning to solve optimization problems via machine learning (ML). Especially for practical applications, learning-based approaches are of great interest because of the high labor costs associated with the development of completely hand-crafted solution approaches. For routing problems such as the traveling salesperson problem (TSP) and the capacitated vehicle routing problem (CVRP), recent ML-based approaches are able to generate good solutions for small problem instances in a fraction of a second (e.g., Kool et al. (2019)). However, in many real-world applications of these problems users gladly accept more computation time for solutions of even higher quality. Recently proposed approaches (e.g., Hottung & Tierney (2020)) address this demand and integrate learning-based components with high-level search procedures. While these approaches offer improved performance over non-search-based methods, they rely on domain knowledge encapsulated in the high-level search procedures.

In this work, we present a learning-based optimization approach for routing problems that is able to perform an extensive search for high-quality solutions. In contrast to other approaches, our method does not rely on domain-specific high-level search procedures. Our approach learns an instance-specific mapping of solutions to a continuous search space that can then be searched via any existing continuous optimization method. We use a conditional variational autoencoder (CVAE) that learns to encode a solution to a given instance as a numerical vector and vice versa. Some genetic algorithm variants (e.g., Gonçalves & Resende (2012)) use numerical vectors to represent solutions to combinatorial optimization problems. However, these approaches rely on decoding schemes that are carefully handcrafted by domain experts. In contrast, our approach learns the problem-specific decoding schema on its own, requiring no domain or optimization knowledge on the side of the user.

The performance of an optimization algorithm heavily depends on the structure of the fitness landscape of the search space, such as its smoothness. If solutions close to each other in the search space are semantically similar, resulting in a smooth landscape, the employed search algorithm can

iteratively move towards the more promising areas of the search space. It has been observed for some problems that variational autoencoders (VAEs) are capable of learning a latent space in which semantically similar inputs are placed in the same region. This allows, for example, a semantically meaningful interpolation between two points in the latent space (see e.g. Berthelot et al. (2018)). However, it is unclear if this property upholds for a conditional latent space that encodes routing problems. We show experimentally that our CVAE-based approach is indeed capable of learning a latent search space in which neighboring solutions have a similar objective function value. Furthermore, we introduce a novel technique that addresses the issue of symmetries in the latent space and show that it enables our method to match and surpass state-of-the-art ML-based methods. We train our method using high-quality solutions because we aim to learn a latent search space that contains mostly high-quality solutions. Hence, our method usually requires a long offline phase (e.g., to generate solutions using a slow, domain-independent, generic solver). However, this offline phase is offset by fast, online solution generation.

We focus on the TSP and the CVRP, which are two of the most well-researched problems in the optimization literature. The TSP is concerned with finding the shortest tour between a set of cities that visits each city exactly once and returns to the starting city. The CVRP describes a routing problem where the routes for multiple vehicles to a set of customers must be planned. All customers have a certain demand of goods and all vehicles have a maximum capacity that they can carry. All routes must start and end at the depot. The task is to find a set of routes with minimal cost so that the demand of all customers is fulfilled and each customer is visited by exactly one vehicle. We consider the versions of the TSP and CVRP where the distance matrix obeys the triangle inequality.

The contributions of this work are as follows:

- We propose a novel approach that learns a continuous, latent search space for routing problems based on CVAEs.
- We show that our approach is able to learn a well-structured latent search space.
- We show that the learned search space enables a standard differential evolution search strategy to outperform state-of-the-art ML methods.

## 2 RELATED WORK

In Hopfield & Tank (1985), it was first proposed to use an ML-based method to solve a routing problem. The authors use a Hopfield network to solve small TSP instances with up to 30 cities. In Vinyals et al. (2015), pointer networks are proposed and trained to solve TSP instances with up to 50 cities using supervised learning. Bello et al. (2016) extend this idea and train a pointer network via actor-critic reinforcement learning. More recently, graph neural networks have been used to solve the TSP, e.g., a graph embedding network in Khalil et al. (2017), a graph attention network in Deudon et al. (2018), or a graph convolutional network in Joshi et al. (2019). The significantly more complex CVRP has first been addressed in Nazari et al. (2018) and Kool et al. (2019), in which a recurrent neural network decoder coupled with an attention mechanism and a graph attention network are used, respectively. While some of these methods use a high-level search procedure (such as beam search), all of them are focused on finding solutions quickly (in under one second). In contrast, our approach is able to exploit a longer runtime (more than one minute for larger instances) to find solutions of better quality.

A couple of approaches use local search like algorithms combined with ML techniques to solve routing problems. Chen & Tian (2019) propose to learn an improvement operator that makes small changes to an existing solution. The operator is applied to a solution iteratively to find a high-quality solutions for the CVRP. However, with a reported runtime of under half a second for the CVRP with 100 nodes, the method is not focused on performing an extensive search. In Hottung & Tierney (2020), another iterative improvement method for the CVRP is proposed that integrates learned heuristics into a large neighborhood search framework. The method is used to perform an extensive search with reported runtimes of over one minute for larger instances. In contrast to our method, the high-level large neighborhood search framework contains domain specific components and is known to perform exceptionally well on routing problems (Ropke & Pisinger, 2006).

Perhaps most similar to our work is the line of research based on Gómez-Bombarelli et al. (2018), in which the authors use a VAE to learn a continuous latent search space for discovering molecules.

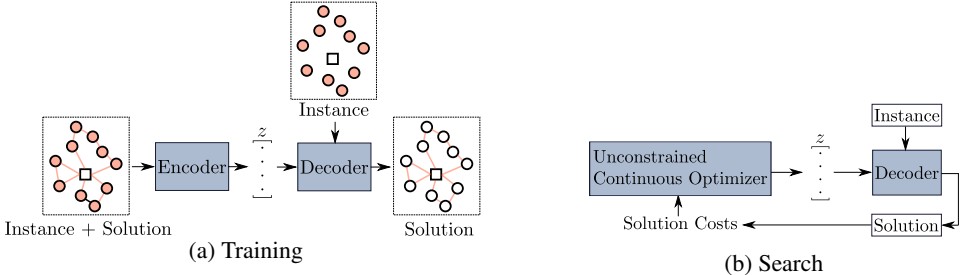

Figure 1: CVAE-Opt overview

They use an additional Gaussian process model that is trained to predict the the quality of molecules given their latent search space representation to allow for a gradient-based search. Kusner et al. (2017) and Jin et al. (2018) use a similar setup, but use Bayesian optimization for the search. Winter et al. (2019) propose to use particle swarm optimization to search a learned latent space for new molecules. To a more limited degree, the idea of optimizing in a continuous learned space has also been used for neural architecture optimization (Luo et al., 2018). In contrast to the aforementioned methods, we do not use a separate model to predict the solution quality based on their latent representation, because decoding and evaluating solutions in our setting is cheap (compared to molecules or neural network architectures). Furthermore, our approach addresses a fundamentally different problem, because routing problems must be solved with respect to a given context (i.e., a problem instance that describes location coordinates that must be visited) and we hence use a CVAE in this work. Learning a latent space conditioned on a problem instance (with the number of possible instances being basically infinite) is significantly more challenging. Ichter et al. (2018) propose to use CAVEs to learn a latent space conditioned on problem instances to represent solutions to robot motion planing problems. However, they only sample solutions at random from the learned distribution and do not perform a guided search. We show that the learned structured latent space of our approach enables a guided search that significantly outperforms random sampling.

Different generative models have been used to sample new population members in probabilistic evolutionary algorithms known as estimation of distribution algorithms (e.g., a Helmholtz machine (Zhang & Shin, 2000), a restricted Boltzmann machine (Tang et al., 2010; Shim et al., 2010; Probst et al., 2017), or a VAE (Garciarena et al., 2018; Bhattacharjee & Gras, 2019)). All these methods are focused on how to explore an existing search space using generative models. In contrast, our method is focused on learning the search space itself, leaving the actual search to a generic optimizer.

## 3 METHOD

Our novel approach, called CVAE-Opt, learns a continuous (latent) search space for routing problems that can be searched by any continuous optimization method. It is based on a CVAE that learns to map solutions to routing problem instances to a continuous, $n$-dimensional space. In contrast to conventional search spaces, the learned latent search is trained to contain only high-quality solutions.

Autoencoders are neural networks that are used to learn an efficient encoding of data. They consist of an encoder and a decoder network. The encoder learns to reduce an input $x$ to a point $z$ in a low dimensional space and the decoder tries to reconstruct the input $x$ based on $z$. The objective of the training is to minimize the difference between the input $x$ and the output of the decoder, requiring the network to learn an efficient encoding of $x$. In contrast, VAEs are generative models that do not use a deterministic encoder, but instead an encoder that parameterizes an approximate posterior distribution over $z$. In our context, we do not want to train the decoder to generate solutions for only a single instance (e.g., a given set of coordinates for the TSP), but instead for all instances of a certain instance type (e.g., all TSP instances with 50 cities). We thus use a CVAE (Sohn et al., 2015), which enables us to learn a latent search space conditioned on the problem instances.

### 3.1 VARIATIONAL AUTOENCODER-BASED COMBINATORIAL OPTIMIZATION

The overall training process of CVAE-Opt is shown in Figure 1a. The stochastic encoder $q(z|l, s)$ receives a problem instance $l$ and a high-quality solution $s$ and outputs an $n$-dimensional vector $z$. The decoder $p(s|l, z)$ is given $z$ together with the instance $l$ and outputs a solution $s'$. One objective

of the training is to minimize the difference between the original high-quality solution $s$ and the solution $s'$ generated by the decoder. While the decoder is powerful enough to construct a good solution based on the instance $l$ alone, it is also given the latent variable $z$ that describes the aspects of the solution $s$ that the decoder cannot reliably infer on its own. The second objective during training is to ensure that high-quality solutions can be generated for values of the latent variable that have not been seen during training. This objective is explained in more detail below.

Figure 1b shows the iterative search process, in which the decoder $p(s|l, z)$ is used together with any unconstrained continuous optimizer to search for solutions to a problem instance $l$. The unconstrained continuous optimizer navigates the search through the learned latent search space. At each iteration, the optimizer outputs a vector $z$ describing a point in the latent search space. The decoder generates a solution $s'$ based on $z$ and the objective function value of $s'$ is returned to the optimizer. With an effective optimizer and the learned search space, high-quality solutions to $l$ can be found.

**Routing problem representation** We describe a routing problem instance by a graph $G = (V, E)$, with $V = \{v_0, ..., v_n\}$. The representation of a problem instance $l$ consists of a set of $n$ feature vectors $x_0, \ldots, x_i, \ldots, x_n$, where $x_i$ describes node $v_i$. For the TSP, each node represents a location (e.g., a city) with each two-dimensional feature vector describing the location's coordinates. For the CVRP, the node $v_0$ represents the depot, and all other nodes represent the customers. As in Nazari et al. (2018), each feature vector is four-dimensional and describes the unfulfilled demand of a location, the remaining capacity of the vehicle, and the coordinates of the location. For both problems, a solution $s$ describes a sequence of locations $v_{s_0}, \ldots, v_{s_T}$ (for the TSP, $T = n$) in which the first location is the starting city (for the TSP) or the depot (for the CVRP). We note that our formalism focuses on routing problems on a Euclidean plane. While we anticipate that our approach will work for other types of combinatorial optimization problems (with adjustment of the input layers), we save showing this for future work.

## 3.2 MODEL

We implement the encoder $q_\phi(z|l, s)$ and the decoder $p_\theta(s|l, z)$ using neural networks, with $\phi$ and $\theta$ denoting the network weights. In earlier work, (e.g., Nazari et al. (2018)) routing problems are often modeled as Markov decision processes where a solution is constructed by a sequence of actions (i.e., which node should be visited next). We follow that approach and train our decoder $p_\theta(s|l, z)$ to select the location that should be added to the solution at each step $t \in \{1, \ldots, T\}$, with the first element of the solution being predefined (TSP: the starting city, CVRP: the depot). As in Nazari et al. (2018), we use a masking schema to prevent the model from selecting actions that would result in an unfeasible solution. The probability of the decoder of generating a solution $s$ can be decomposed as (Sutskever et al., 2014):

$$p_\theta(s|l, z) = \prod_{t=1}^{T} p(s_t|s_0, \ldots, s_{t-1}; l; z). \tag{1}$$

Like the decoder, the encoder generates the latent variable $z$ for a solution $s_0, \ldots, s_T$ to a problem instance $l$ sequentially. At step $t \in \{1, \ldots, T\}$ it encodes the $t$-th element of the solution.

Similar to Nazari et al. (2018), we allow the input representation to change during encoding and decoding. The input $x_{0,t}, \ldots, x_{n,t}$ at time step $t$ can be changed to reflect the new sub-problem defined by the problem instance $l$ and the constraints introduced by the partially constructed solution $s_0, \ldots, s_{t-1}$. In the following, we omit the index $t$ when referring to the input data of the model to allow for better readability. For the CVRP, we update the demands of the customers and the remaining vehicle capacity based on the decisions of the model in earlier decoding steps. For the TSP, we make no changes to the problem instance representation.

**Network architecture** The architecture of the encoder and the decoder is shown in Figure 2. Both use a linear embedding layer and an attention mechanism to encode/decode solutions sequentially. Weights are shared between identical components in the encoder and decoder. This allows not only for faster training, but also enforces a shared view of the encoder and decoder on the given problem representation. A more detailed description of the network architecture is given in Appendix A.

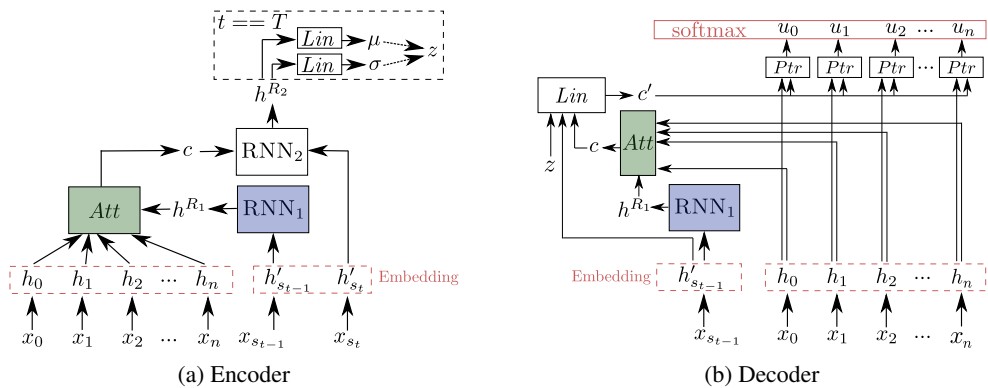

(a) Encoder

(b) Decoder

Figure 2: CVAE-Opt model

## 3.3 TRAINING

The objective of the model training is two-fold. The first objective is to maximize the (log-)likelihood of reconstructing a solution $s$ to an instance $l$ encoded by the encoder $q_\phi(z|l, s)$ via the decoder $p_\theta(s|l, z)$. The second objective is to keep the posterior distribution of the encoder close to a given desired probability distribution $p(z)$. We use a standard Gaussian distribution ($\mu = 0$, $\sigma = 1$) for $p(z)$ and measure the difference between both distributions with the Kullback–Leibler (KL) divergence. As in $\beta-$VAEs (Higgins et al., 2017), we weight the objectives during training using the parameter $\beta$:

$$\mathcal{L}(\phi, \theta, s, l, z, \beta) = \mathbb{E}_{q_\phi(z|l,s)}[\log p_\theta(s|l, z)] - \beta \, D_{KL}(q_\phi(z|l, s)||p(z)). \qquad (2)$$

**Symmetry breaking** Optimization problems are commonly subject to *symmetrical solutions*, which are multiple solutions that represent the same semantic solution, but differ in terms of their syntax. For example, for the TSP the solution sequence $s_0, \ldots, s_n$ represents the same solution as the solution sequence $s_1, \ldots, s_n, s_0$. For the CVRP, the subtours can be ordered in any order in the solution sequence without changing the underlying solution. This might lead to the CVAE placing identical solutions in different regions of the learned latent search space because they are represented by different solution sequences. To force the model to learn a representation of the underlying solution and not the solution sequence, we train the model to reproduce a symmetrical solution to the input, rather than the exact same solution as the input. The symmetrical solutions used during training are chosen at random for each epoch.

## 4 EXPERIMENTS

We evaluate CVAE-Opt on datasets of TSP and CVRP instances and compare it to state-of-the-art optimization approaches. We use two different unconstrained continuous optimizers in our experiments: a basic differential evolution (DE) algorithm (Storn & Price, 1997) and random search (RS). In the following, we refer to the two variants of CVAE-Opt as CVAE-Opt-DE and CVAE-Opt-RS. In all experiments, CVAE-Opt is run on a single Nvidia Tesla V100 GPU and a single core of a Intel Xeon 4114 CPU at 2.2 GHz[1]. We evaluate CVAE-Opt on TSP and CVRP instances with 20, 50, and 100 nodes. For each of these six problem classes, we generate instances with identical properties to the instances used in Kool et al. (2019) using the instance generator made available by the authors. We use 93,440 instances for model training, 100 for search validation, and 1,000 for testing the search per problem class.

### 4.1 SETUP

**Training** For the TSP, we solve all instances to optimality using CONCORDE (Applegate et al., 2006). For the CVRP, we create high-quality solutions using the heuristic solver LKH3 (Helsgaun, 2017). We run LKH3 a single time for each instance with the hyperparameter configuration used in Helsgaun (2017). We train separate models for each instance class for 300 epochs. Every 25 epochs

---

[1]Our implementation of CVAE-Opt is available at `https://github.com/ahottung/CVAE-Opt`

the model is evaluated by using its decoder in a CVAE-Opt search setting to look for solutions to the 100 validation instances. The search setup (i.e., the hyperparameter configuration) is identical to the one used in the later testing/deployment stage. The model offering the best validation performance is used to search for solutions to the test instances.

The ideal selection of the hyperparameter $\beta$ depends on the problem class and the search setup. For each problem class we repeat the training process a small number ($<20$) of times and pick the model with the best validation search performance. All other search hyperparameters are identical over all training runs and have not been tuned. The training batch size is set to 128 and the Adam optimizer (Kingma & Ba, 2014) with a learning rate of $10^{-3}$ is used.

**Search** The DE algorithm employed in CVAE-Opt-DE maintains a population of vectors in the learned latent search space that is improved by crossover and mutation. Offspring vectors are created by combining three vectors of the population using vector arithmetic, as described in Storn & Price (1997). We slightly modify the employed DE algorithm to better profit from the parallel computing capabilities of a GPU: Instead of generating one offspring solution at a time, we decode and evaluate *a batch* of solutions per iteration.

In all experiments of CVAE-Opt-DE, we use a DE population size of 600. At each iteration of the DE algorithm, 600 offspring vectors are generated and decoded in one batch. The initial population vectors are sampled uniformly at random from the bounded search space. To determine the bounds, we encode 1,000 separate model validation instances (with the encoder) to points in the latent space. The bounds are then selected so that 99% of the coordinates of the points are within the bounds. This ensures that the search operates in regions of the latent search space known by the decoder even if the posterior distribution of the encoder differs substantially from the standard Gaussian distribution. The *crossover probability CR* and the *differential weight F* of the DE are set to 0.95 and 0.3, respectively. Solutions are generated greedily by the decoder (i.e., the action with the highest probability value is selected at each step). The search terminates after 300 iterations. We note that we do not tune these hyperparameters and that the reported results can thus likely be improved.

In CVAE-Opt-RS, the latent variables are sampled randomly from a Gaussian distribution. We also evaluated sampling from a uniform distribution using the same bounds as for CVAE-Opt-DE, but observed that this slightly deteriorates the performance. All other components of CVAE-Opt-RS (and its hyperparameters) are identical to CVAE-Opt-DE.

## 4.2 Symmetry breaking

First, we evaluate the effectiveness of our symmetry breaking mechanism. We train five models with symmetry breaking and five models without symmetry breaking for the TSP and the CVRP with 50 and 100 nodes each. In all training runs $\beta$ is set to 1e-3. Figure 3 shows the search performance of the models after the final training epoch on the validation instances (in terms of the gap to the solutions obtained via Concorde and LKH3). In all cases our symmetry breaking mechanism leads to an significant performance improvement. For the TSP instances the mean gap is reduced from 0.09% to 0.04% for instances with 50 nodes, and from 1.23% to 0.37% for instances with 100 nodes. Similarly, for the CVRP symmetry breaking reduces the mean gap from 3.18% to 0.33% and 5.66% to 1.67% for instances with 50 and 100 nodes, respectively.

## 4.3 Influence of $\beta$

To evaluate the influence of the parameter $\beta$, we repeat the training with different $\beta$ values (again five times per value and problem setting). We only consider TSP and CVRP instances with 100 nodes because the experiments are computationally expensive. Figure 4 shows the performance (gap to Concorde and LKH3) of the models after the final training epoch when searching for solutions to the validation instances. We observe that in our setting the best search performance can be observed for $\beta$ values of 1e-2 and 1e-3 for the TSP and the CVRP, respectively. This is a significant deviation from the proposed $\beta$ values ($> 1$) in Higgins et al. (2017). A high $\beta$ value corresponds to a strong limit on the capacity of the latent information channel. We hypothesize that our extensive search procedure benefits from a latent (search) space that is able to represent many instances.

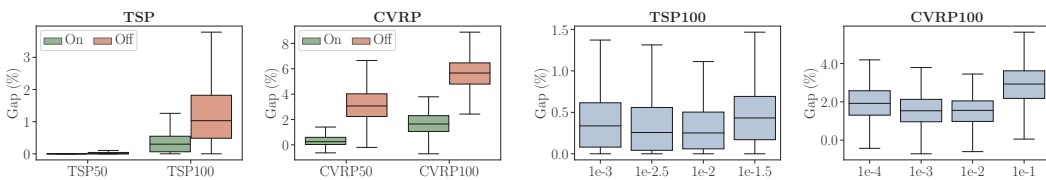

Figure 3: Symmetry breaking          Figure 4: Influence of $\beta$

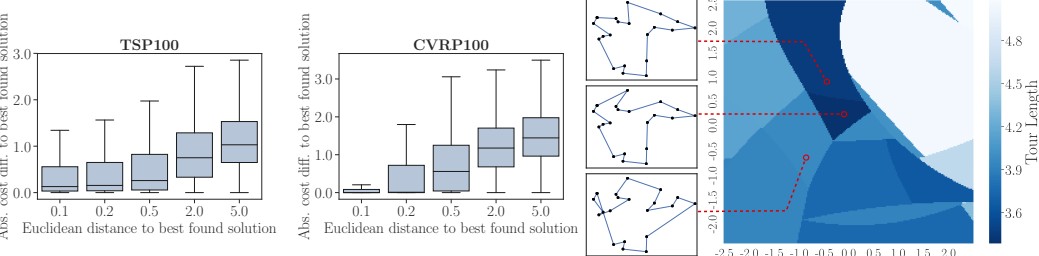

Figure 5: Solution quality vs. distance in the latent search space

Figure 6: Learned latent search space for the TSP along 2 of 100 dimensions

## 4.4 STRUCTURE OF THE LEARNED SEARCH SPACE

The performance of any search algorithm depends on the structure of the search space. Ideally, solutions of similar quality should be placed in similar regions of the search space. We conduct the following experiment to evaluate if our method learns a (latent) search space in which solutions of high quality can on average be found in the proximity of other high-quality solutions: First, we sample 1,000 solutions for a routing problem instance from the learned search space. The best of these solutions functions as a reference solution. Next, we sample solutions from multiple hyperspheres around the reference solutions, only considering points within the defined bounds of the search space. For each hypersphere we sample 100 solutions, and discard all solutions that are identical to the reference solutions. We repeat this experiment for each of the 1,000 test instances per problem class. Figure 5 shows the absolute cost difference of the sampled solutions to the reference solution for the TSP and CVRP with 100 nodes (see Appendix B for all results). The results show for all problem classes that, on average, solutions close to the high-quality solutions are also of similar quality (in contrast to solutions farther away), indicating that the search space is well structured.

This experiment also shows that our method successfully learns a search space mostly containing high-quality solutions. Even randomly selected solutions that have a euclidean distance of five from the high-quality reference solution only have an average absolute cost difference of 1.13 for the TSP and 1.51 for the CVRP (both with 100 nodes). As an illustrative example, Figure 6 shows of a learned latent search space for randomly selected TSP instance with 20 nodes (the search space is only shown along 2 of 100 dimensions). While this visualization is not artificially selected, it does not allow for any generalizable assertions.

## 4.5 COMPARATIVE EXPERIMENTS

**TSP**   For a comparison to the state-of-the-art, we compare CVAE-Opt-DE and CVAE-Opt-RS to the AM approach from Kool et al. (2019). We run the AM approach on the same machine as CVAE-Opt using the code and the models made available by the authors, sampling 500,000 solutions for each instance. Figure 7 shows the performance for all three methods over the course of the search process (with a 95% confidence interval). For instances with 20 nodes all methods achieve a very low ($< 0.1$) average gap to optimality, albeit the AM method performs slightly worse than the CVAE-based approaches. Instances with 50 and 100 nodes are computationally harder and allow CVAE-Opt-DE to take advantage of its guided search in the learned latent search space. For both instance groups, CVAE-Opt-DE outperforms the AM approach and CVAE-Opt-RS after the first few seconds of the search. This is the case although CVAE-Opt-DE needs significantly more time per sampled solution than the other approaches. Table 1 shows the final results after the completion of the search and additionally compares the performance of CVAE-Opt to Concorde, LKH3 and the

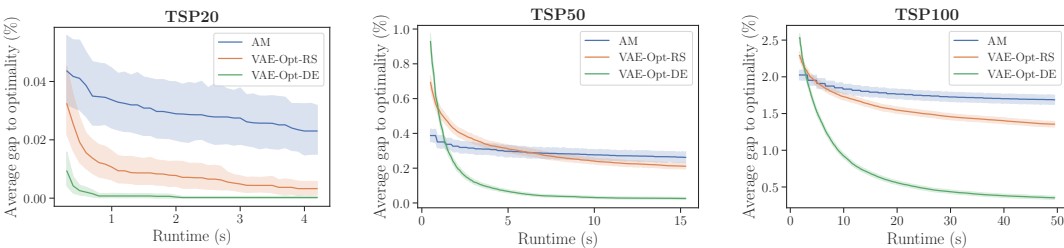

Figure 7: TSP search performance (gap to optimality)

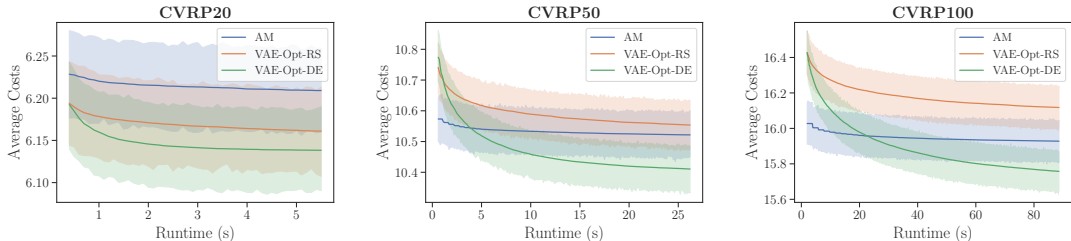

Figure 8: CVRP search performance (absolute costs)

graph convolutional network approach using beam search and the shortest-tour heuristic (GCN-BS) from Joshi et al. (2019). We note that GCN-BS, in contrast to other evaluated learning-based methods, solves instances in batches (of size 200) making a direct comparison of the runtime difficult.

**CVRP** First, we compare CVAE-Opt-DE and CVAE-Opt-RS to the AM approach using the same hyperparameters as for the TSP instances. Figure 8 shows the performance of all three methods. Note that we report the absolute cost instead of the gap to optimality because it is not currently computationally feasible to solve our CVRP instances to optimality. For all three instance sizes CVAE-Opt-DE outperforms both other methods given similar runtime. We note that the significant performance difference between CVAE-Opt-RS and CVAE-Opt-DE is the most unbiased confirmation that our approach is able to learn a well-structured search space. If the learned search would have no meaningful structure, we would expect both approaches to have similar performance.

Table 1 shows additional results comparing both CVAE-Opt implementations to LKH3, NLNS (Hottung & Tierney, 2020), and NeuRewriter (Chen & Tian, 2019). We run all approaches except for NeuRewriter on the same machine as CVAE-Opt. For NLNS, we use 10 cores and limit the runtime to the time needed by CVAE-Opt-DE. For NeuRewriter, we report the results obtained by the authors (we thus mark the results with a star) on instances with identical properties. CVAE-Opt-DE finds better solutions than NeuRewriter on all instance sizes (due to its much longer runtime) and comes close to the performance of LKH3 and NLNS (which profit from expert designed high-level search components that CVAE-Opt does not require) on instances with 20 and 50 customers.

## 4.6 GENERALIZATION

We evaluate the generalization performance of CVAE-Opt-DE and the AM approach by using a model trained on instances with 100 nodes to solve instances with 95, 105, 125 and 150 nodes. We

Table 1: Comparison to existing approaches on TSP and CVRP instances

| | TSP | | | | | | CVRP | | | | | |
| --- | --- | --- | --- | --- | --- | --- | --- | --- | --- | --- | --- | --- |
| | Avg. Gap | | | Runtime | | | Avg. Cost | | | Runtime | | |
| Method | n=20 | n=50 | n=100 | n=20 | n=50 | n=100 | n=20 | n=50 | n=100 | n=20 | n=50 | n=100 |
| CVAE-Opt-DE | 0.00% | 0.02% | 0.34% | 10.5 | 21.5 | 55.1 | 6.14 | 10.40 | 15.75 | 20.8 | 41.0 | 94.7 |
| CVAE-Opt-RS | 0.00% | 0.20% | 1.35% | 6.3 | 16.8 | 50.5 | 6.15 | 10.54 | 16.11 | 16.6 | 38.2 | 92.5 |
| AM | 0.02% | 0.25% | 1.65% | 4.3 | 21.7 | 82.1 | 6.21 | 10.52 | 15.93 | 6.2 | 27.3 | 97.2 |
| GCN-BS | 0.00% | 0.02% | 1.25% | 0.1 | 0.2 | 0.3 | - | - | - | - | - | - |
| NeuRewriter* | - | - | - | - | - | - | 6.16 | 10.51 | 16.10 | 0.1 | 0.2 | 0.4 |
| NLNS | - | - | - | - | - | - | 6.14 | 10.36 | 15.54 | 22.7 | 40.8 | 93.3 |
| Concorde | 0.00% | 0.00% | 0.00% | 0.1 | 0.2 | 0.5 | - | - | - | - | - | - |
| LKH3 | 0.00% | 0.00% | 0.00% | 0.1 | 0.6 | 2.8 | 6.14 | 10.36 | 15.54 | 7.9 | 29.0 | 53.8 |

Table 2: Generalization ability of models trained on instances with 100 nodes

| | TSP | | | | | CVRP | | | | |
| | Avg. Gap | | Runtime | | | Avg. Gap to LKH3 | | Runtime | | |
| n | CVAE-Opt | AM | VAE-Opt | AM | Conc. | CVAE-Opt | AM | CVAE-Opt | AM | LKH3 |
|---|---|---|---|---|---|---|---|---|---|---|
| 95 | 0.31% | 1.60% | 55.9 | 73.8 | 0.4 | 1.27% | 2.37% | 92.6 | 85.8 | 52.0 |
| **100** | 0.34% | 1.65% | 55.1 | 82.1 | 0.5 | 1.36% | 2.46% | 94.7 | 97.2 | 53.8 |
| 105 | 0.41% | 1.72% | 61.5 | 92.0 | 0.5 | 1.39% | 2.41% | 101.5 | 105.3 | 58.8 |
| 125 | 0.74% | 2.04% | 74.7 | 130.5 | 0.7 | 2.08% | 2.77% | 128.4 | 149.6 | 68.3 |
| 150 | 1.45% | 2.74% | 107.6 | 185.8 | 1.0 | 3.24% | 3.69% | 166.6 | 209.4 | 73.1 |

Table 3: Comparison to DE using a handcrafted continuous decoder

| | TSP | | | | | | CVRP | | | | | |
| | Avg. Gap | | | Runtime | | | Avg. Cost | | | Runtime | | |
| Method | n=20 | n=50 | n=100 | n=20 | n=50 | n=100 | n=20 | n=50 | n=100 | n=20 | n=50 | n=100 |
|---|---|---|---|---|---|---|---|---|---|---|---|---|
| CVAE-Opt-DE | 0.00% | 0.02% | 0.34% | 10.5 | 21.5 | 55.1 | 6.14 | 10.40 | 15.75 | 20.8 | 41.0 | 94.7 |
| Opt-DE | 2.36% | 14.06% | 32.28% | 11.0 | 22.0 | 56.0 | 6.32 | 12.18 | 23.97 | 21.1 | 41.1 | 95.2 |

mainly focus on the ability to generalize to larger instances, because using a model trained on small instances to tackle large-scale problems could be a viable option if training on large-scale instances is too computationally expensive. The results are shown in Table 2. Note that for the CVRP, we report the gap to LKH3 to allow for better comparability of the results over the different instance sizes. For TSP and CVRP instances with 95, 100 and 105 nodes there is no notable performance difference, which shows the ability of our model to generalize well to instances that are slightly different than the training instances. This is an important aspect for the application of our method in practice. For instances with 125 and 150 nodes the performance is significantly worse. We note that impaired performance on instances that differ substantially from the instances seen during training is to be expected. However, this does not severely limit the applicability of our method because there are many scenarios in which the distribution of encountered instances does not change frequently.

## 4.7 ABLATION STUDY

We replace the learned decoding schema in CVAE-Opt-DE with a handcrafted decoder from the literature to further evaluate to what extent learning plays a role in CVAE-Opt's performance. Opt-DE implements the decoding schema proposed by Bean (1994) while adopting all other components of our learning-based method. The decoder of Opt-DE takes in a vector $z \in [0-1]^n$ that defines a permutation of the $n$ nodes of a problem instance, which is constructed by sorting the nodes according to their corresponding entry in $z$, i.e., node $v_i$ corresponds to entry $z_i$. A tour is constructed by trying to visit the nodes in the order of the permutation. For the CVRP we use the same masking schema as for CVAE-Opt to avoid illegal tours. We limit the search time of Opt-DE to the time needed by CVAE-Opt-DE and note that the handcrafted decoder is significantly faster than the learned decoder. The results are shown in Table 3. CVAE-Opt-DE outperforms Opt-DE on the TSP and CVRP for all instance sizes, with the difference being especially visible on larger problems.

## 5 CONCLUSION

We presented CVAE-Opt, a method that uses a variational autoencoder to learn a mapping of routing problem solutions to points in a continuous (latent) search space. The learned space can be searched by any basic unconstrained continuous optimizer. The approach provides an interface between optimization and machine learning techniques, allowing traditional continuous optimization methods to search in a learned space. We show that our approach is able to learn a well-structured search space that enables a guided search by a high-level, domain independent continuous optimizer. On TSP and CVRP instances, CVAE-Opt significantly outperforms state-of-the-art ML-based approaches. In the future, we will further investigate the properties of the learned search space and evaluate recent extensions to the VAE framework.

ACKNOWLEDGMENTS

The computational experiments in this work have been performed using the Bielefeld GPU Cluster.

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

# A  NETWORK ARCHITECTURE DETAILS

## A.1  ENCODER

The encoder encodes a routing problem solution $s_0, \ldots, s_T$ sequentially. The model is given the input features of the nodes $x_{s_{t-1}}$ and $x_{s_t}$ at decoding step $t \in \{1, \ldots, T\}$ separately from the problem representation of all nodes $x_0, \ldots, x_n$. For each of the inputs of the problem representation $x_0, \ldots, x_n$, an embedding $h_i$ is created using a linear transformation that is applied to all inputs separately and identically. For the separate input features of the nodes $x_{s_{t-1}}$ and $x_{s_t}$, a different linear transformation is applied in a similar fashion to generate the embeddings $h'_{s_{t-1}}$ and $h'_{s_t}$. All learned embeddings have a dimensionality of $d_h$, and we set $d_h$ to 128 for all trained models.

The first recurrent neural network module $RNN_1$ receives the embedding $h'_{s_{t-1}}$ of the previously visited node in the solution $s$ at each step $t$. The output $h^{R_1}$ contains information on the first $t-1$ elements of the solution. We implement all recurrent neural networks in the model as gated recurrent neural networks (Chung et al., 2014).

All embeddings are used by the attention layer $Att$ to compute a single $d_h$-dimensional context vector $c$ that describes all relevant embeddings $h_0, \ldots, h_n$. The relevance of each input is determined based on the current encoding state given by $h^{R_1}$. To compute the context vector $c$, first the $n$-dimensional alignment vector $\bar{a}$ is computed that describes the relevance of each input:

$$\bar{a} = softmax(u_0^H, ..., u_n^H), \tag{3}$$

where

$$u_i^H = z^A \tanh(W^A[h_i; h^{R_1}]). \tag{4}$$

Here, $z^A$ is a vector and $W^A$ is a matrix with trainable parameters and ";" is used to describe the concatenation of two vectors. Based on the alignment vector $\bar{a}$, the context vector $c$ is generated:

$$c = \sum_{i=0}^{n} \bar{a}_i h_i. \tag{5}$$

The context vector $c$ is then used by the recurrent neural network module $RNN_2$, which is the main encoding component of the encoder. At each step $t$ it is given the embedding of the $t$-th node in the solution sequence $x_{s_1}, \ldots, x_{s_T}$ in addition to $c$. Its output $h^{R_2}$ in the last iteration $T$ encodes the complete sequence $x_{s_1}, \ldots, x_{s_T}$ and is used in two separate linear transformations to calculate the $d_h$-dimensional vectors $\mu$ and $\sigma$. These vectors parameterize a multivariate normal distribution from which the latent variable $z$ is sampled using the reparameterization trick (Kingma & Welling, 2014).

## A.2  DECODER

The architecture of the decoder is based on the model proposed in Nazari et al. (2018). At each step $t$ the model uses a pointer mechanism (Vinyals et al., 2015) to point towards the node that should be visited next. The decoder uses the same embedding, attention mechanism, and recurrent neural network $RNN_1$ as the encoder. The weights for these components are shared by the encoder and decoder. In addition to the inputs required to calculate $c$, the decoder also gets the latent variable $z$ as an input. The concatenation $[z, c, x_{s_{t-1}}]$ is transformed by a linear layer to a $d_h$-dimensional vector $c'$. This vector provides the context to the pointing mechanism that calculates the output distribution over all actions based on the node embedding embedding $h_0, \ldots, h_n$:

$$p_\theta(a_t|\pi_t) = softmax(u_0, ..., u_n), \tag{6}$$

where

$$u_i = z^B \tanh(W^B[h_i; c']), \tag{7}$$

and the vector $z^B$ and the matrix $W^B$ contain trainable parameters.

# B  SEARCH SPACE STRUCTURE ANALYSIS FOR ALL PROBLEM SIZES

Figure 9 shows the absolute cost difference and the euclidean distance of the sampled solutions to the reference solution (i.e., the best solution found in a random search of 1,000 solutions) for all problem classes.

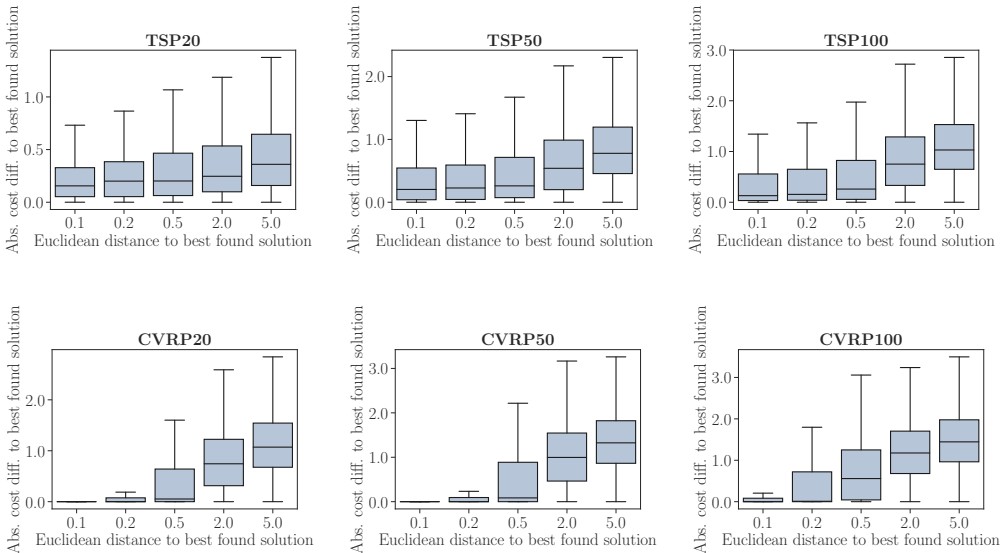

Figure 9: Solution quality vs. distance in the latent search space

