# OpenReview forum: "Learning a Latent Search Space for Routing Problems using Variational Autoencoders"
_ICLR.cc/2021/Conference — ICLR 2021 Poster_

### Official Review · AnonReviewer1 · 2020-10-26
**Interesting combination of ideas, but lacks motivation and novelty**

**Rating:** 5
**Confidence:** 4

**Review:**

$\textbf{Summary.}$ This paper proposes a new approach to solve routing problems based on (1) training a variational autoencoder (VAE) to model the optimal solution of problems (i.e., supervised learning) and (2) applying continuous search methods for decoding at test time. While I think the idea of this paper is promising, it is limited in the sense that (1) the method is not particularly novel, (2) the VAE model relies on supervised learning for training, (3) the experiments are limited to small-scale settings, and (4) the proposed method does not show any promise over the "traditional" solvers, e.g., Concorde or LKH3.

$\textbf{Pros.}$
- I think the idea of transforming the combinatorial search problem into a continuous search problem is interesting. This direction has not been considered in the literature of using DNNs for combinatorial optimization.
- The proposed framework can easily be generalized to combinatorial optimization problems other then routing problems, e.g., minimum vertex cover.

$\textbf{Cons.}$
- The proposed framework is not particularly novel when compared to the existing works on de novo drug design using DNNs (which is also a combinatorial optimization). The paper mentions how routing problems are different from de novo drug design tasks since they require the DNN-solver to encode context of the given problem. However, this does not make the problem much more challenging and the paper simply uses conditional VAE to take care of the context encoding.
- Importantly, the proposed framework use supervised learning for training the model and requires extracting optimal solutions from the training graphs using existing solvers. This severely limits applicability of the proposed framework. In comparison, (Kouter et al. 2019) trained the AM model using reinforcement learning to motivate DNNs for solving combinatorial problems where exact solvers do not exist.
- For small scale instances (considered in the experiments), exact solvers can usually find solution better than the heuristics. I encourage the authors to provide more results on large-scale experiments, since this is where non-exact solvers are needed. This is especially significant since it is computationally intractable to extract optimal solutions from large-scale problems. How would the proposed method handle this case?
- I think the proposed framework lacks motivation, since the proposed method does not show much promise over the traditional solvers such as Concorde or LKH3. This is especially significant since the proposed framework requires using traditional solvers to extract optimal solutions on the training graphs. In what situations would users be encouraged to use the proposed framework?
- In the experiments, I think the authors should add a baseline proposed by (Lu et al. 2020) since they also considered using neural networks to solve the routing problems. While they use certain domain-specific components, I think the proposed framework also relies on domain knowledge (since they require exact solvers for training).
- I suggest adding a reference to (Winter et al. 2019), who proposed to use swarm optimization on the latent space of VAE for molecular optimization.

$\textbf{References.}$

(Winter et al. 2019) Efficient Multi-Objective Molecular Optimization in a Continuous Latent Space, Chemical science 2019

(Lu et al. 2020) A Learning-based Iterative Method for Solving Vehicle Routing Problems, ICLR 2020

---

> ### Author Response · Authors · 2020-11-21
> **Response (2/2)**
>
> 4. Performance vs ML-based methods: Our method is not able to outperform state-of-the-art optimization methods like Concorde and LKH3, but neither is any other ML-based method that does not contain domain-specific, handcrafted components. However, the gap between learned methods and expert-designed operations research (OR) methods is steadily closing, and we hope that you agree that our research contributes towards that.
> The potential of a method that learns heuristics (for problems for which no handcrafted heuristics exist) is immense. Consider an example from our recent research. Without piercing the double blind barrier, we worked with a company to route technicians to make repairs to ships. The routing problem they face has side constraints that make it unsolvable with Concorde or LKH3, and modifying these two solvers to handle it is also not trivial. We modeled the problem with a mixed-integer program, meaning we can find solutions, but often not optimal ones, and finding optimal solutions takes a long time. The company needs a heuristic (in the OR sense) to quickly find solutions, but lacks the OR competency to write one. Our technique could be applied in such cases, even if that entails a computational expensive offline phase to generate training solutions.
> 5. Additional baseline/reference: We find the approach of Lu et al. (2020) interesting and impressive. However, it is built on existing traditional heuristic methods for routing problems. We pursue a very different goal: Learning a heuristic method without requiring heuristic problem knowledge on the side of the user. While our method requires existing high-quality solutions, these solutions can be generated (albeit very slowly) without heuristic knowledge (e.g., using a mixed-integer solver). For example, for routing problems mixed-integer formulations already exist and can be used to generate training solutions. These solutions can then be used to learn a heuristic that finds solutions quickly by exploiting the properties of distribution of instances at hand. Regarding the suggested reference to Winter et al. (2019): We thank you for making us aware of this paper and we will cite it in the final version of our paper.

---

> > ### Comment · AnonReviewer1 · 2020-11-22
> > **Thank you for the response.**
> >
> > Thank you for the detailed and thoughtful response, especially given the limited time.
> >
> > After reading your response, I found that there are certain details that I missed out during reading the paper. Hence, I am lowering my confidence to from 5 to 4. I also found the arguments made by authors to be reasonable. Hence, I am raising my score from 4 to 5. I did not raise my score to 6 since the authors did not provide a solid (empirical) evidence for their arguments.
> >
> > 1. Novelty
> >
> > I think your response regarding novelty is convincing, and I now think the novelty of this paper is above the bar of ICLR 2020.
> >
> > 2. Supervised learning
> >
> > I apologize for missing to see that you use LKH3 to extract solutions for training the model. However, my main concerns are rather about using "highly optimized solvers" for extracting the solution, i.e., the proposed algorithm requires domain knowledge enough to build a "highly optimized solver" for extracting the solutions. While one may use the standard solvers like Gurobi and CPLEX (or even simple algorithms like 2-opt local search), I do not know whether if such a plan works to generate meaningful results in practice.
> >
> > 3. Scale of instances
> >
> > I think your response regarding the scale of instances are quite reasonable. However, it would still be nice to see the results on large-scale instances because the methods may simply become too slow for generating solutions (since the model generates each route sequentially). However, using the size 150 may not be considered large-scale for CVRP or TSP problems, e.g., CVRPLIB benchmark (http://vrp.atd-lab.inf.puc-rio.br/index.php/en/) considers problems more than 10000 vertices).
> >
> > 4. Performance vs ML-based methods
> >
> > I agree with the authors that ML-based methods do not have to outperform the state-of-the-art optimization methods right away. However, I believe the empirical standards should become higher as time goes. I really like the example provided by authors, i.e., a combinatorial optimization problem that (a) does not have a domain-specific solver, (b) can be modeled using a mixed-integer solver, and (c) solved effectively using your algorithm. However, I am not sure if I could take this into account, since there are no solid examples or results in the paper.
> >
> > 5. Additional baselines/reference (regarding comparison with Lu et al. (2020))
> >
> > Similar to my previous concern, I am not sure whether if generating solutions without heuristic knowledge (e.g., using a mixed-integer solver) can generate meaningful outputs in practice. I would encourage the authors to evaluate their algorithms on problems where methods like Lu et al. (2020) cannot be applied.

---

> > > ### Author Response · Authors · 2020-11-24
> > > **Response**
> > >
> > > Thank you very much for your quick response, we do not take it for granted. Thank you for acknowledging the novelty of our approach and increasing your score. We carefully read your response and we think that all your remaining concerns are mainly based on 1) your (justified) doubt that the necessary training data can be generated in a reasonable timeframe and 2) your concern the considered instances are too small and that our method might not scale to larger instances. Please let us address these two points:
> > >
> > > 1. First, we would like to acknowledge that exact, generic solvers are very complex programs that sometimes take forever to solve an arguably very simple problem, while solving other complex problems in the blink of an eye. We hope you agree that there are some problems for which no fast exact solver exists and for which an exact solver takes more than a few minutes, but less than a few hours to generate high-quality solutions. These are the cases in which the necessary training data for our method can be generated and our method can be used to speed up solution generation. Is our method a silver bullet that can be applied to any routing problem in the industry? Certainly not, because - as you correctly point out - in some cases the training solution generation might be too computationally expensive. Does our method advance the research focused on learning heuristic-based solution approaches? We think so, and there are definitely plenty of problems where it can be applied.
> > > 2. You mention that the well-known CVRPLIB contains instances with more than 10,000 nodes. However, we want to point out that only 5 of the 263 instances in the CVRPLIB contain more than 10,000 nodes. These instances are basically outliers and the majority of research on the CVRP is still focused on much smaller instances. The CVRP instances we present with 100 customers are not considered large-scale, but they are still relevant in the industry today and difficult to solve. We would really like to evaluate our method on larger instances, but alas, we do not have the computational budget to do so.  We can only mention that there is no component that should not scale to instances with more than 1,000 nodes given the computational resources (see also our discussion of computationally complexity in our response to reviewer 4) and note that if a company or group wanted to build a VAE-Opt heuristic on such a dataset, it would definitely be possible; but the computation time required to perform research (ablation, etc. on large problem sizes) is just too much for us. We hope that our work contributes towards the research on ML-based heuristics despite our computational constraints (for example, by releasing our source code so that others can easily use our method on larger instances).

---

> ### Author Response · Authors · 2020-11-21
> **Response (1/2)**
>
> Thank you for your valuable feedback! We are excited that you find the idea of searching a learned continuous search space for solutions to combinatorial optimization problems interesting and that you agree that it has not yet been considered in the literature. We are also pleased that you concur with us that our framework can (easily) be applied to other optimization problems. In the following we will address your concerns and hope that we can convince you that this paper deserves to be accepted at ICLR.
>
> 1. Novelty: While you agree that the “direction [of our paper] has not been considered in the literature of using DNNs for combinatorial optimization”, you argue that the idea is not particularly novel, because we only used an existing method and applied it (without making any significant changes) to a new problem. While you acknowledge that solutions to combinatorial optimization problems need to be encoded/decoded with respect to the problem instance, you write that this “does not make the problem much more challenging and the paper simply uses conditional VAE to take care of the context encoding”. While we agree that the idea of using a conditional VAE to tackle a conditional encoding/decoding problem is somewhat straightforward, we strongly disagree that this does not make the problem much more challenging. Learning a conditioned latent space is more challenging and there was no indication that this idea would work well enough to learn a (well-structured!) search space that allows for an iterative search. Furthermore, our approach does not perform well out of the box: we show that the beta parameter needs to be selected carefully (learning does work successfully using only the default VAE loss function) and propose a symmetry breaking mechanism that significantly improves performance. Please also see our discussion of novelty in the response to all reviewers at the top.
> 2. Supervised learning: You write that the “proposed framework use[s] supervised learning for training the model and requires extracting optimal solutions” which would limit the applicability of our method. First, we would like to note that our method does not require optimal solutions for training! In fact, we do not train our CVRP models using optimal solutions, because of the difficulty to find these optimal solutions. Instead we use high-quality solutions generated with the heuristic solver LKH3, and note that solutions could also be found with a standard solver like Gurobi or CPLEX. This sort of modeling is fairly easy to do, but is often too slow to put into practice. Our technique allows to trade offline training time (including the time to generate training solutions) for faster online solution generation.
> On a more general note, we agree with you that using reinforcement learning for the training of our method would be ideal. However, learning a conditional latent search space is already a very hard task and using reinforcement learning would make this even harder. We do intend to explore this idea in the future.
> 3. Scale of instances: We agree that the used TSP instances are trivial to solve. However, we report results for them because the TSP is the standard benchmark for ML-based routing approaches. We disagree that the used CVRP instances are small-scale and that optimal solutions can easily be found. In fact, the Gurobi solver, considered state-of-the-art for general mixed-integer programs, needs more than 1 hour to solve CVRP instances with only 50 customers (hence we use non-optimal solutions for training). We hope that this can convince you that the considered CVRP instances are already hard to solve and that our method has no problem with instances for which optimal solutions are not available. Additionally, we are going to add results for even larger instances to the final version of the paper (available soon), which show the generalization ability of models trained on n=100 to instances with 150 nodes.

---

### Official Review · AnonReviewer4 · 2020-10-27
**Great work!**

**Rating:** 7
**Confidence:** 2

**Review:**

Overall great work and highly readable, as reviewer I am leaning to accept

##########################################################################

Summary:


The paper presents a way of solving routing problems (which are discrete and highly combinatorial)
by optimising the input to an instance-specific mapping from a continuous vector space to the space of solutions for that particular instance (desiring high objective value solutions).
The mapping is obtained from masked autoregressive sampling with a decoder trained as part of a conditional VAE.

##########################################################################

Reasons for score:


Overall, I vote for accepting. The idea presented is neat and seems to work well in the instance size range presented . I have one major concern about generalisation in problem-instance sizes (expanded below in Cons 1). The paper is readable. The experiments could be improved and I make suggestions for that. Hopefully the authors can address my concern in the rebuttal period.


##########################################################################Pros:

Pros:

1. The paper leverages a re-parametrization of the search space to make search better and in this case to even allow unconstrained continuous optimization methods for the problem of routing in a way that successfully generalizes to unseen instances (the best solutions to routing problems are problem-specific).


2. The paper is clear and readable.


##########################################################################

Cons:


I have not taken into account points 2-5 for making review score decisions, but I believe they should be addressed in a camera-ready worthy version of the paper


1. It isn't clear to me if the model is trained (and tested) on instances of varying sizes or fixed size. Generalising/extrapolating to unseen problem-instance sizes is really important.
  + If the approach is size-specific then any unseen problem-instance size at test time will require generating new data and training, which would limit the impact of the approach.
  + If the approach isn't size specific (from what I understood in the Appendix A it isn't) what is the training instance size distribution? From the section 4 wording I understood that the sizes are fixed, but I could have misunderstood.
    - One possibility for showing size-generalisation/extrapolation could be training on one size (e.g. n=50) and testing on the other two sizes (e.g. n=20 and n=100). (Edit: the authors have done this)

2. There is the implicit claim that learning helps, but there is no ablation/comparison to a randomly initialised model, I expect it to do really badly, but even in that case it would only help to show that it is the learning/training component (and not the architecture's inductive biases alone) that make the approach successful. Running CVAE-Opt-DE with a randomly initialised model and adding that to Table 1 would be great. (Edit: the authors have addressed this)
3. The authors show their approach can be used by any unconstrained continuous optimization algorithm by evaluating one and a variation of it. There is the possibly implicit claim of "any optimization algorithm can be used and still get great results" and it might merit having optimization method different-enough from the other two. (Edit: the authors have explained this)
4. The current decoding takes O(solution_size * problem_size) time, as the problem-instance is passed at each time-step, this will limit the scalability to large input sizes. (Edit: the authors have clarified this is fine in the subdomain they are targeting)
5. It is not clear to me if Figure 5 shows one representative per equivalence class, or possibly more than one. (Edit: the authors have clarified)


##########################################################################

Random thoughts

Being able to function-approximate the decoding plus measuring objective value, would allow gradient-based planning, and make the approach even better, Figure 5 suggest we might be able to do a linear/quadratic function of the latent search space points to the objective values (but even then variance could be too high). Maybe a cheaper alternative to know if this is worthwhile is plotting for each pair of solutions in Figure 5 the distance in x-axis and the cost-diff in y-axis (a per-solution view instead of being histogram aggregated)

##########################################################################

Questions during rebuttal period:

Please address and clarify the cons above

---

> ### Author Response · Authors · 2020-11-24
> **Response**
>
> Thank you for your valuable and positive feedback! We sincerely apologize that we are responding to you so late in the discussion phase. We are very excited that you like our paper and that you voted to accept it. Based on your feedback, we performed additional experiments and updated the paper. We hope that you like the new version even better. Please let us discuss how we address your remaining concerns in the updated version of the paper.
>
> 1. Generalization: You understood the original version correctly in that the instance sizes were fixed for each model in the evaluation. In response to your concern, we performed additional experiments that evaluate the generalization performance of our method (see Section 4.6 in the paper). The experiments show that our method can indeed be applied to instances that are slightly different than the training instances without a significant performance impact. This allows for an applicability of our method in scenarios in which the distribution of encountered instances does not change frequently.
> 2. Ablation study: Your wish for an ablation study that disables the learning component of our method is similar to the request from reviewer 3. We are happy to report that we can provide the requested experiment. Please see Section 4.7 in the updated version of our paper. In the experiments we replace the learned decoder with a handcrafted decoder from the literature. The experimental setup is very similar to your suggested setup (using an untrained network) but without the actual network. This drastically improves performance (in terms of runtime) and removes the randomness introduced by the random weight initializations. Furthermore, in our setup the optimal solution is guaranteed to be in the continuous search space. We can reasonably assume that the network architecture does not have an inductive bias, because the outputs for all nodes are calculated using identical network weights (i.e., the same operation is applied to each input separately and identically).
> 3. Implicit claim that our method works well with any optimizer: We evaluate our method with two optimizers: random search (RS) and differential evolution (DE) and show that both optimizers are able to find solutions with a small gap to optimality. However, DE performs significantly better than RS. We would like to argue that the majority of optimizers should perform at least as well as RS (assuming reasonable hyper-parameters), meaning that the majority of optimizers should be able to find solutions with a small gap to optimality. This does not mean that the majority of available optimizers performs as well as DE, but we also do not claim that in the paper. Consider that the black box optimization community has a wealth of solvers available (e.g., CMA-ES, DE variants, particle swarm variants), and none dominates any other across all problem types.  To summarize, we implicitly claim that any optimizer works with our method (as well as RS), but not that any optimizer works as well as DE.
> 4. Computational complexity: You are correct. The computational complexity of our approach is basically O(n^2). However, if you look at the runtime of our approach (including the results for the generalization performance) you will notice that the runtime does not grow quadratically. This is because of the parallel computing capabilities of the used GPU, which allows it to perform operations in parallel for instances up to a certain size (even for CVPR instances with 150 customers our model needs only 900MB of GPU memory at test time).  Note that the considered CVPR instances are already extremely difficult to solve to optimality and that solution approaches for CVRP instances with <= 100 customers are still actively researched by the operations research community. In fact, 40% of the instances contained in the CVRPLIB (which are frequently used as a basis for comparison in the literature) have <= 100 customers, meaning that our approach can solve instances of relevant sizes in a reasonable timeframe.
> 5. Figure 5: Yes, the figure shows more than one representative per instance and distance. Each graph shows the results over 1,000 test instances and each box is based on up to 100 solutions per instance.
> 6. Gradient-based optimization: We already discuss gradient optimization in our response to reviewer 4 (see the last point in our response). While Figure 5 might suggest that a linear or quadratic function might suffice to map points from the space to the objective values, we can also tell you based on our experience made in the experiments that the variance is indeed too high. While on average, solutions close to the high-quality solutions are also of similar quality (in contrast to solutions further away), there are many solutions for which that is not the case. We think that more complex functions are required. We leave that for future work (including a more detailed analysis of distance vs. cost in the learned space).

---

### Official Review · AnonReviewer2 · 2020-10-28
**Solid approach, empirical evaluations can be strengthened**

**Rating:** 7
**Confidence:** 4

**Review:**

This paper proposes a method to learn a continuous latent space via CVAE to represent solutions to routing problems. Combined with differentiable evolution search algorithms, one can search in the learned latent space for solutions to new problem instances at test time. The proposed method is evaluated on two classes of routing problems: TSP and CVRP. Results show better performance in terms of objective values and runtime. They are also competitive with established expert-designed algorithms such as LKH3.

Positives:
1. The proposed method is novel. The specific algorithm to realize the idea of learning continuous latent spaces is sensible and easy to understand, thanks to clear writing.

2. A continuous latent space enables one to use existing tools in continuous optimization, as this paper demonstrates via two examples: differentiable evolution and random search.

3. The evaluations on two practically relevant routing problems demonstrate the advantage of the proposed method. The experimental section includes useful insight into hyperparameter selection and interpretations of learned latent spaces.

Negatives:
1. Comparison with Joshi et al. (2019) should be included for TSP as well, especially their beam search variants. Given a major benefit of the proposed method is to enable search, comparing with beam search can better evaluate the practical advantage of the proposed method. The sampling procedure used in AM by Wool et al. (2019) improves solution quality but is not a direct search procedure.

2. The problem size is relatively small with the largest problems having 100 nodes. I expect the proposed method, with the resulting continuous latent space, achieves larger improvements for larger problem instances. Can the authors consider larger scale experiments?

Question:
1. For symmetric breaking, does the symmetrical solution to each input problem change between epochs?

2. Is it possible to employ gradient-based optimization algorithms in the learned latent space?

=====================
Post-rebuttal comments:

Thank the authors for the detailed responses and revised submission. My concerns have been adequately addressed and I raised my score to 7.

---

> ### Author Response · Authors · 2020-11-23
> **Response**
>
> Thank you for your valuable feedback! We apologize that we are responding to you so late in the discussion phase. We have focused on updating the paper (please also see our comment at the top) as per your suggestions. We are excited that you point out the novelty of our approach and that you highlight the useful insights of our experiments. We hope that you will agree with us that the updated version of our paper is more than only marginally above the acceptance threshold. Please let us explain how we address your concerns in the updated version of our paper.
> 1. Additional baseline: We now compare our approach to the beam search variant of Joshi et al. (2019). We use the code and the models provided by the authors to run the experiments on the machines that we used for all other experiments. Note that their approach solves instances in batches (to utilize GPU memory) and is much faster than CVAE-Opt and the AM approach, which both solve instances sequentially. We tried to lower the batch size and to increase the beam width size to allow for a more fair comparison, but we observed a worsening performance, presumably because the employed batch normalization is less stable for lower batch sizes. Note also that while both batch and sequential solving are interesting from a research perspective, we consider sequential solving somewhat more realistic from an industrial perspective.
> 2. Scale of instances: First, we would like to note that - while TSP instances with 100 nodes are considered trivial - CVRP instances with 100 customers are not easily solvable to optimality. In fact, the Gurobi solver, considered state-of-the-art for general mixed-integer programs, needs more than 1 hour to solve CVRP instances with only 50 customers. Instances with <= 100 customers are also still used to evaluate state-of-the-art operations research solvers. 40% of the instances contained in the CVRPLIB (which are frequently used as a basis for comparison) have <= 100 customers.
> Nonetheless,  we understand your wish for even larger instances. As a response, we evaluated generalization performance of our model on larger instances with 150 customers. If you compare the runtime of our method, you will notice that it scales almost linearly with the instance size indicating that our method can be applied to larger problems. While we would like to provide you with results for models trained on even larger instances, we note that our computational resources are always limited. However, there is no component of our system that should not scale to CVRP instances with 500 or even 1000 customers given enough computational resources. Furthermore, while our resources are limited, the costs of training a model for large instances should be significantly outweighed by potential saving in an industry setting.
> 3. Question regarding symmetry breaking: Yes, new symmetrical solutions are generated at random for each epoch. We updated the paper to make this more clear.
> 4. Question regarding gradient-based optimization: Yes, in general it should be possible to use gradient-based optimization. For example, we could train another neural network that predicts the quality of a solution (i.e., the objective function value) based on their latent representation z (similar to Gomez-Bombarelli et al. (2018)) and use that to calculate the gradients. However, gradient-based techniques will inevitably get stuck in an optimum, since the latent space comes with no structural guarantees. To escape these local optima a high-level metaheuristic or a heuristic component like momentum would be needed. Furthermore, it is unclear how well the additional neural network could predict solution quality in our conditional setting! Ultimately, we decided to first evaluate the most obvious method to explore a large continuous space (an unconstrained continuous optimizer) and to leave the rest for future work.

---

### Official Review · AnonReviewer3 · 2020-10-28
**Borderline paper**

**Rating:** 6
**Confidence:** 3

**Review:**

The authors propose an algorithm for routing problems by (1) using conditional variational autoencoder to learn a latent space for solutions, and (2) performing black-box continuous optimization such as evolutionary algorithms in the space. The proposed method outperforms existing neural-based methods on a benchmark of the traveling salesmen, and the capacitated vehicle routing problem.

The presentation and writing of the paper are excellent. I really like the intro and related work section where the authors have done a comprehensive survey and position their work among existing ones.

Although the method is very neat and general, the idea of learning a latent space for solutions to combinatorial search problems is not new. It has been investigated in both continuous control\cite{ichter2018learning} and drug design\cite{gomez2018automatic}. In particular, \cite{ichter2018learning} also uses a conditional variational autoencoder to address the issue brought by different problem instances. The authors should discuss more on the challenges of the current problem, why previous approaches cannot apply here, and focus on how to deal with the challenges, otherwise, the contribution will be weak.

Also in the experiment section, it would help the readers understand the approach better by including ablation studies such as applying evolutionary and random search methods directly in the solution space. There is no need to design sophisticated mutation strategies that involve domain knowledge. Simple heuristics will suffice here.

In general, I think this is a borderline paper mostly due to its weak contributions. I will consider raising the score if the authors address the issues well.

Additional question: what is the problem distribution? how do you generate them?



@inproceedings{ichter2018learning,
  title={Learning sampling distributions for robot motion planning},
  author={Ichter, Brian and Harrison, James and Pavone, Marco},
  booktitle={2018 IEEE International Conference on Robotics and Automation (ICRA)},
  pages={7087--7094},
  year={2018},
  organization={IEEE}
}


@article{gomez2018automatic,
  title={Automatic chemical design using a data-driven continuous representation of molecules},
  author={G{\'o}mez-Bombarelli, Rafael and Wei, Jennifer N and Duvenaud, David and Hern{\'a}ndez-Lobato, Jos{\'e} Miguel and S{\'a}nchez-Lengeling, Benjam{\'\i}n and Sheberla, Dennis and Aguilera-Iparraguirre, Jorge and Hirzel, Timothy D and Adams, Ryan P and Aspuru-Guzik, Al{\'a}n},
  journal={ACS central science},
  volume={4},
  number={2},
  pages={268--276},
  year={2018},
  publisher={ACS Publications}
}


Update after rebuttal:
Thanks for the response! I think it resolves my concerns on novelty and evaluation. Hence I raise my score to 6.

---

> ### Author Response · Authors · 2020-11-23
> **Response**
>
> Thank you for your valuable feedback! We apologize that we are responding to you so late in the discussion phase. We have focused on updating the paper (which now includes new experiments) as per your suggestions. Furthermore, we are thrilled that you already like the presentation and writing of the paper (especially the related work section) and that you already like the overall idea of our method. Please let us explain how we address your concerns in the updated version of our paper.
> 1. Novelty: As with reviewer 1 you were not convinced of the novelty of our method, and we acknowledge that we did not make the novelty of our method clear enough in the original version of the paper. We hence updated the introduction and the related work section to make the novelty of our method more clear. Please see our response to all reviewers (that has been especially written with your review in mind) and the first part of our response to reviewer 1. We thank you for making us aware of the work of Ichter et al., which we now discuss in the related work section of our paper.
> 2. Ablation study: Reviewer 4 also requests an ablation study that removes the learning component from our method and replaces it with something handcrafted (you note that simple heuristics will suffice). First, let us note that simple heuristics in the context of discrete optimization are hard to come by, as one must always make problem-specific design choices. However, we have designed an experimental setup to remove the learned component from our method while still keeping all other components (like the differential evolutionary algorithm and the masking schema) in place: We replace the learned decoder with the handcrafted decoder from Bean (1994) (introduced in the context of random-key genetic algorithms). The decoder takes in a vector z \in [0-1]^n (with n being the number of nodes), which defines a permutation of the nodes by sorting the nodes by the value in the decoder for each node. A tour is constructed by trying to visit the nodes in the order of the permutation. For the TSP, the decoder is basically identical to the argsort(z) operation. For the CVPR we use the same masking schema as for VAE-Opt to avoid that illegal tours are constructed. While this decoding procedure is simple (and thus fulfills the requirement that it can be developed without much domain-knowledge) it has proven to be effective (e.g., see Snyder et al. (2006)). VAE-Opt outperforms the decoder on both problems and all sizes of each problem.  Please see Section 4.7 and Table 2 in the paper for further details.
> 3. Problem distribution: To allow for comparability of the results we use the same problem distribution as in other ML-based papers (e.g. Nazari et al., Kool et al., and Chen at al.) that address the TSP and the CVRP. In fact, we use the instance generator made available by Kool et al. to generate the instances. For the TSP, the locations are sampled uniformly at random in the unit square. For the CVRP the demand of the customers is sampled uniformly from {1,...,9} and the vehicle capacity is set to 30, 40, 50 for instances with 20, 50, 100 customers, respectively. We now make it more clear in the paper that we use the instance generator from Kool et al.
>
>
> References:
>
> Bean, J. C. (1994). Genetic algorithms and random keys for sequencing and optimization. ORSA journal on computing, 6(2), 154-160.
>
> Snyder, L. V., & Daskin, M. S. (2006). A random-key genetic algorithm for the generalized traveling salesman problem. European journal of operational research, 174(1), 38-53.

---

### Author Response · Authors · 2020-11-20
**Response to all reviewers**

First, we would like to thank everybody for reviewing our paper and for your valuable comments. Most reviewers asked for additional experiments and we are currently waiting for these experiments to finish. We apologize that we won’t be able to provide you with the results before next week and hope that you understand that performing experiments on a shared infrastructure sometimes takes longer than expected (to ensure the validity and comparability of the results we conduct the experiments in the same setting as our earlier experiments).

We will address each of your comments/suggestions individually at a later point. In this response we want to focus on addressing an issue that is independent of the outcome of the additional experiments, namely novelty. While R2 states that the “method is novel” and R4 does not raise any concerns regarding novelty, R3 states that “idea of learning a latent space for solutions to combinatorial search problems is not new” and R2 states that the method “is not particularly novel”.

The novelty of our approach stems from the fact that routing problems (and many operations research problems) have a different structure than, e.g., drug design or motion planning, and is two-fold:
1. We show that the *combination* of (a) searching a continuous latent learned space for discrete solutions and (b) using a CVAE to condition the latent space on different problem instances is necessary for solving routing problems. Note that Ichter et. al. only sample from the space, an approach we show is less effective than search on the TSP and CVRP. Furthermore, it cannot be simply assumed that the learned space has a structure that can be searched. We show experimentally that our approach is indeed able to learn a conditioned latent space in which semantically similar inputs are placed in the same region. Finally, we want to emphasize that routing problems have a different structure (spatial costs, permutation-based space, and in the case of the CVRP, a vehicle “state” for the capacity), that separate it from the drug-design and path planning problems.
2. We introduce a symmetry breaking mechanism that, while simple, is rather effective at improving the method’s performance. This mechanism is domain specific, but the idea is a general one, and could be effective in creating latent spaces for other optimization problems. Indeed, symmetry breaking is widespread in methods for discrete optimization problems, but it has yet to be used when learning to optimize.

Note that we will make this more clear in the updated version of our paper, as well as cite Ichter et al.

---

### Author Response · Authors · 2020-11-23
**We uploaded an improved version of the paper.**

We updated the paper as per the suggestions of the reviewers:
1. We improved the introduction and the related work section to better highlight the novelty of our method and our contribution.
2. We provide an ablation study to evaluate to what extent learning plays a role in VAE-Opt’s performance as requested by reviewer 3 and reviewer 4.
3. We evaluate the generalization performance of our method (especially on instances that are larger than the training instances).
4. We compare to the method proposed by Joshi et al. (2019).

Furthermore, we made other minor changes to address questions and comments of the reviewers.

---

### Decision · Program_Chairs · 2021-01-07
**Final Decision**

**Decision:**

Accept (Poster)

**Comment:**

This paper proposes a learning based approach for solving combinatorial optimization problems such as routine using continuous optimizers. The key idea is to learn a continuous latent space via conditional VAE to represent solutions and perform search in this latent space for new problems at the test-time. The approach is novel and experiments showed good results including ablation analysis.

Reviewer comments are adequately addressed during the response phase and I find the changes satisfactory. Overall, this is a good paper and I recommend accepting it.

One last comment: It would be a great addition if the paper could add discussion about the applicability of this approach to arbitrary combinatorial optimization problems and what design choices are critical to come up with an effective instantiation.